# Analysis of Climate Change in the Caucasus Region: End of the 20th–Beginning of the 21st Century

**Alla A. Tashilova [1,\*], Boris A. Ashabokov [1,2], Lara A. Kesheva [1] and Nataliya V. Teunova [1]**

[1] Federal State Budgetary Institution «High-Mountain Geophysical Institute», Lenin av., 2, 360030 Nalchik, Russia; ashabokov.boris@mail.ru (B.A.A.); kesheva.lara@yandex.ru (L.A.K.); n.teunova@gmail.com (N.V.T.)

[2] Institute of Computer Science and Problems of Regional Management, Kabardino-Balkarian Research Center, Russian Academy of Sciences, I. Armand str., 37a, 360000 Nalchik, Russia

\* Correspondence: tashilovaa@mail.ru

**Abstract:** The study of climate, in such a diverse climatic region as the Caucasus, is necessary in order to evaluate the influence of local factors on the formation of temperature and precipitation regimes in its various climatic zones. This study is based on the instrumental data (temperatures and precipitation) from 20 weather stations, located on the territory of the Caucasian region during 1961–2011. Mathematical statistics, trend analysis, and rescaled range Methods were used. It was found that the warming trend prevailed in all climatic zones, it intensified since the beginning of global warming (since 1976), while the changes in precipitation were not so unidirectional. The maximum warming was observed in the summer (on average by 0.3 °C/10 years) in all climatic zones. Persistence trends were investigated using the Hurst exponent $H$ (range of variation 0–1), which showed a higher trend persistence of annual mean temperature changes ($H = 0.8$) compared to annual sum precipitations ($H = 0.64$). Spatial-correlation analysis performed for precipitations and temperatures showed a rapid decrease in the correlation between precipitations at various weather stations from $R = 1$ to $R = 0.5$, on a distance scale from 0 to 200 km. In contrast to precipitation, a high correlation ($R = 1.0$–$0.7$) was observed between regional weather stations temperatures at a distance scale from 0 to 1000 km, which indicates synchronous temperature changes in all climatic zones (unlike precipitation).

**Keywords:** temperature; precipitations; warming; Hurst exponent; persistence; spatial correlation; Caucasian region

## 1. Introduction

The problem of climate change is extremely urgent today. The global climate on our planet is changing rapidly. In this regard, an increasing number of studies are being devoted to this problem [1–17]. The Russia territory is more sensitive to the effects of climate than the Northern Hemisphere and the rest of the globe. Throughout Russia, the average growth rate of average annual air temperature has been 0.46 °C/10 years in 1976–2017. This is 2.5 times the growth rate of global temperature over the same period: 0.18 °C/10 years, and more than 1.5 times the average warming rate of surface air over the Earth's land: 0.28 °C/10 years (estimates according to the Hadley Center and the University of East Anglia: HADCRUT4, CRUTEM4) [18].

Observations of regional climate show that atmospheric phenomena are more significant and variable in regional rather than globally. Many factors affect the climatic features of the south of Russia, including zonal and altitudinal zonality. The geographic position in temperate latitudes contributes to the formation of a moderately continental type of climate, while the Caucasus Mountains serve as a climate cliff between the temperate and subtropical zones. The Caucasus region has significant impact

on climatic features, which is supported by air masses, bringing the Mediterranean warm moist air. An important factor is the difference in altitudes from the Caspian lowland (−28 m from sea level) to the peaks of the North Caucasus, with the highest point in Europe—Mount Elbrus with a height of 5642 m. According to the nature of the relief, the North Caucasus is usually divided into three zones: the plain (Black Sea zone, steppe, Caspian zone), with a height above sea level (a.s.l.) of less than 500 m a.s.l.; foothills (500–1000 m a.s.l.); mountain (>1000 m a.s.l.) and high-mountain (>2000 m a.s.l.). The issues of climate change in areas of the National Park "Prielbrusye" (the high-mountain zone) and the Sochi National Park (the Black Sea zone) are especially important, since they can be beyond landscapes causing disturbance of the ecosystem balance [19–25].

An important aspect of this region is an assessment of the regional response of the mountain climate against the backdrop of global warming, to study the glaciers deglaciation mechanisms. As research continues into climate of the Caucasus region, it becomes apparent that unfortunately, the historical information about climate fluctuations in the high mountains of the Caucasus is very scarce and not systematized. Due to the lack of long-term observations in mountainous areas, some authors [26] restored to the meteorological regime of the corresponding area according to NCEP/NCAR reanalysis, and corrected the information using data from individual instrumental observations. Others [27] restored to the series of temperatures and precipitation at the meteorogical stations Teberda (1280 m a.s.l., Teberda state biosphere reserve) and Terskol (2144 m a.s.l., Elbrus national park) of the Caucasus region, using dendroclimatological methods.

In the first case [26], for the restored meteorological regime in the Caucasus from 1948 to 2013, fragmentary observation materials of the first Elbrus expedition in July–August 1934–1935 [28], and of the second Elbrus expedition in 1957–1959, and in 1961–1962 (Institute of Geography, USSR Academy of Sciences) [29] were used. Analysis of the recovered data for the period 1948–2013 have shown that in the area of Mount Elbrus during the warm season, the positive trend does not go beyond the limits of natural variability, and the change in the average annual temperature is characterized as a stable value "−0.01 °C/10 years".

According to the results from the dendrological analysis [28], it was concluded that the mountain landscapes of Teberda and Terskol in 1960–2005 was characterized by relatively stable climatic conditions. In general, in this area there is a tendency to a slight increase in the air temperature in individual months, and to an increase in annual precipitation. However, under the conditions of an extremely rare network of meteorological observations, it is difficult to reliably determine the causes of this phenomenon, and attribute the differences due to the influence of local factors or the lack of representativeness of weather stations.

The series thus restored have one significant drawback for the study, they cover different time intervals, and the statistical characteristics of such meteorological series cannot be compared to each other. In addition, reanalysis of the data obtained using satellite meteorology can lead to heterogeneity of the series until the mid 70s.

The only way to somewhat reliably estimate regional climate change is by statistical analysis of long series of instrumental data covering the same time span, such as the approach in this paper.

The report from the Intergovernmental Panel on Climate Change (IPCC) of 31 March 2014 states that there are more significant climate changes on all continents, and spaces [30]. The observed effects of climate change have affected ecosystems of land and ocean, some sources of human livelihoods, water supply systems, agriculture, and human health. In this context, the study of climate and the identification of its possible consequences, have now become scientific problems that attract great attention from researchers around the world.

## 2. Materials and Methods

The focus of our study was climate of the Caucasian region (southern Russia), whose territory in the context of the article was limited to 41.28–47.14 degrees north latitude (°N) and 38.58–48.17 degrees east longitude (°E).

To study the climate in different regions of southern Russia, we used data from meteorological instrumental observations (1961–2011) by 20 weather stations, of the state observational network of Roshydromet and provided by the North Caucasian Administration for Hydrometeorology and Environmental Monitoring. (Table 1 and Figure 1). The data of the time series were homogeneous, throughout the period under study the location of the stations remained constant (outside populated areas), and the so-called urban warming did not affect them. The average, maximum, and minimum seasonal and annual temperatures in the south of Russia were investigated.

**Table 1.** Geographical location of weather stations inside the Caucasian region.

| № *n/n* | Weather Stations | Longitude (°N), Latitude (°E) | Height above the Sea Level, m (m a.s.l.) |
|---|---|---|---|
| | | Plain stations (<500 m a.s.l.) | |
| 1 | Sochi (Black Sea zone) | 43.35° N; 39.73° E | 57 |
| 2 | Krasnodar (steppe zone) | 45.20° N; 38.58° E | 26 |
| 3 | Izobil'nyi (steppe zone) | 45.22° N; 32.42° E | 194 |
| 4 | Mozdok (steppe zone) | 43.44° N; 44.39° E | 126 |
| 5 | Prokhladnaya (steppe zone) | 43.46° N; 44.05° E | 198 |
| 6 | Rostov-on-Don (steppe zone) | 47.14° N; 39.44° E | 64 |
| 7 | Maykop (steppe zone) | 44.37° N; 40.05° E | 270 |
| 8 | Derbent (Caspian zone) | 42.04° N; 48.17° E | 30 |
| 9 | Kizlyar (Caspian zone) | 43.51° N; 46.43° E | −17 |
| 10 | Makhachkala (Caspian zone) | 42.59° N; 47.31° E | 173 |
| 11 | Izberg (Caspian zone) | 42.34° N; 47.45° E | 21 |
| | | Foothill stations (500–1000 m a.s.l.) | |
| 12 | Stavropol | 45.03° N; 41.58° E | 540 |
| 13 | Cherkessk | 44.17° N; 42.04° E | 526 |
| 14 | Kislovodsk | 43.54° N; 42.43° E | 819 |
| 15 | Nalchik | 43.22° N; 43.24° E | 500 |
| 16 | Vladikavkaz | 43.21° N; 44.40° E | 680 |
| 17 | Buinaksk | 42.49° N; 47.07° E | 560 |
| | | Mountain stations (1000–2000 m a.s.l.) | |
| 18 | Teberda | 43.45° N; 41.73° E | 1280 |
| 19 | Akhty | 41.28° N; 47.44° E | 1054 |
| | | High-mountain station (>2000 m a.s.l.) | |
| 20 | Terskol | 43.15° N; 42.30° E | 2144 |

In previous studies [31,32], trends in the amount of precipitation and daily maximums of precipitation were analyzed in the Caucasus region, an analysis of the temperature regime was added in this study. In the series of temperatures, averaged values, anomalies (deviations of the observed value from the norm), and trends for the four seasons and the calendar year (January–December) were considered. The climatic norm was considered to be the mean multi-year value of the considered climate variable for the base period of 1961–1990 [33]. The anomalies were calculated for each year as the difference between the current value and the norm of the corresponding climate variable (average 1961–1990). In the series of mean temperature and sum precipitation, the data were averaged within the calendar seasons of each year (the winter season included December of the previous year) and for the year as a whole. Maximum and minimum temperatures were defined as the largest and lowest

values for a certain period (month). The absolute maximum (minimum) was the largest (smallest) value was observed at least once in a month. We used absolute maxima and minima for each month of the season during the period 1961–2011.

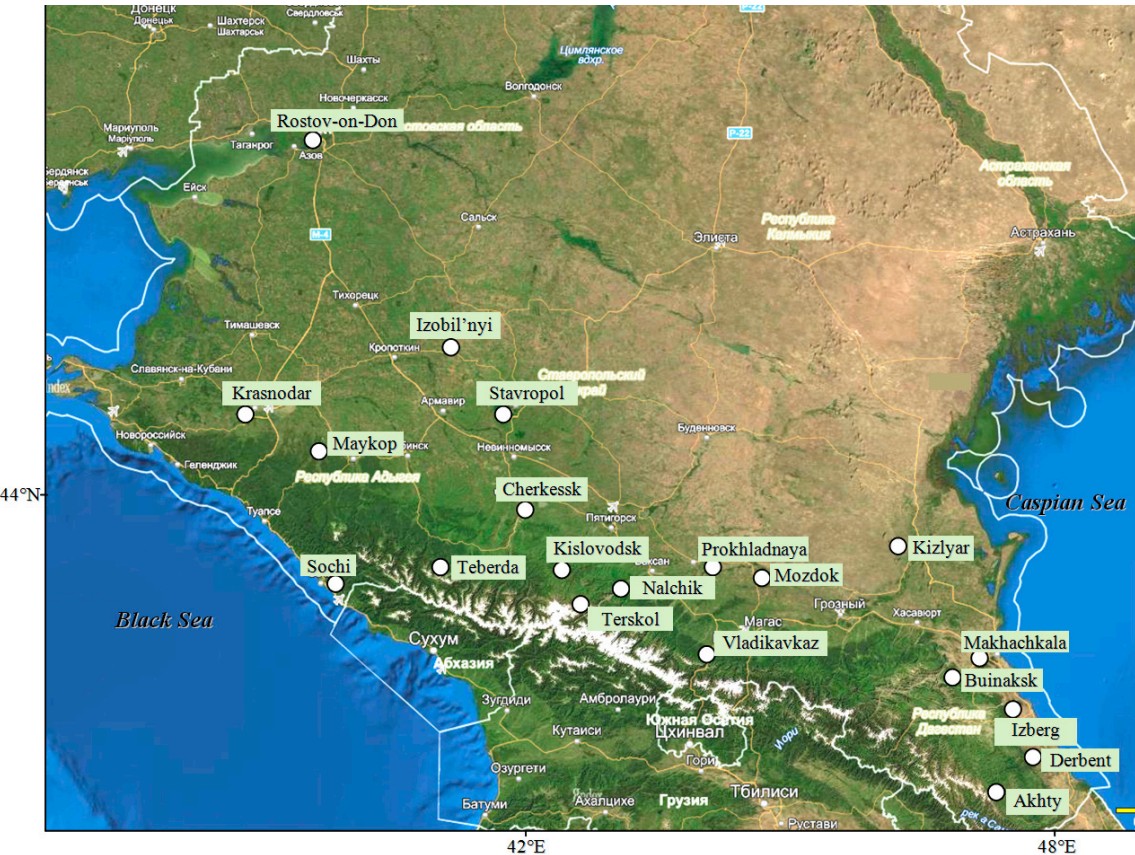

**Figure 1.** Geographical location of weather stations inside the Caucasian region.

Time series were investigated by statistical methods, as well as by means of the STATISTICA, SPSS 15.0 programs [34–36], spatial fields of distributions were constructed using the geoinformation system Golden Software Surfer 8 [37]. Linear trends characterizing the trend of the considered value over the entire observation period from 1961 to 2011, and from 1976 (the beginning of global warming) were built in Excel. The estimation of the linear trend coefficients was considered the least squares method degrees per decade, °C/10 years.

To accept the hypothesis regarding the presence of a statistically significant linear trend, a 95% significance level ($\alpha$) was adopted and determined through a determination coefficient $R^2$ characterizing the share of the trend in the explained variance (*D*, %). Using the coefficient of determination $R^2$, it was possible to check the significance of the trend. For this, the F-criterion was determined:

$$F = \frac{R^2}{1 - R^2} \frac{n - k - 1}{k},\tag{1}$$

where *k* is the number of trend equation coefficients. The constructed linear regression trend was significant with a level of significance $\alpha$, if the inequality held:

$$F > F_{1-\alpha;k;n-k-1}\tag{2}$$

Quantile $F_{1-\alpha;k;n-k-1}$ calculated in Excel by expressing FINV($\alpha; k; n - k - 1$).

The lower threshold value of the coefficient of determination, which determines the statistical significance of the trend at a 95% confidence interval, was *D* = 8% (for *n* = 51, 1961–2011).

Climate of the Caucasus region, which includes various climatic zones (Table 1 and Figure 1), was primarily determined by its position in the temperate latitudinal climate zone.

The North Caucasus mountain system prevents the movement of cold air masses from north to south, and warm from south-west and west to north-east and east. A complex local circulation is created in the mountains with the separation of the two temperate zone regions: the Atlantic-continental (plain, foothill) and the mountain (high-mountain). At the same time, atmospheric processes in the region are complicated by local factors, namely the complex orography of the North Caucasus. Due to the complexity of climate formation in such a complex orographically heterogeneous terrain, the correlation of meteorological parameters of stations located in different climatic zones was of interest.

The spatial structures of the air temperature fields and precipitation fields were analyzed from weather station data in different climatic zones, and spatial correlation relationships between them were determined depending on the scale of the distance between them.

The study assessed the persistence of climate change. As its integral characteristic, the rescaled range method ($R/S$ analysis) and fractal properties of time series (Hurst exponent $H$), were used [38–44]. By using the rescaled range method for the first time, the British hydrologist Harold Hurst studied the rise of the Nile River, as well as the sequence of measurements of atmospheric temperature, rainfall, river flow parameters, thickness of annual wood growth rings, and other natural processes [38]. The method is based on the analysis of the range $R$ of the meteorological parameter (the largest and smallest value in the segment under study), and the standard deviation $S$ and its dependence on the period of the studied time $t$. The Hurst exponent can distinguish a random series from a nonrandom one, even if the random series is not Gaussian (that is, not normally distributed).

To calibrate the time series, Hurst introduced a dimensionless ratio by dividing the range $R$ by the standard deviation $S$ of the observations. The range of $R_n$ is the difference between the maximum and minimum levels of accumulated deviation $X_n$.

$$R_n = \max\left(X_k - \frac{k}{n}X_n\right) - \min\left(X_k - \frac{k}{n}X_n\right), \tag{3}$$

where $X_n = x_1 + x_2 + \ldots + x_n$, $n \geq 1$;

$X_n$—accumulated deviation in n steps ($t$ periods);
$R_n$—deviation range in $n$ steps, where

$S_n = \sqrt{\dfrac{\sum\limits_{k=1}^{n} x_k - \overline{x}^2{}_n}{n}}$—empirical standard deviation;

$\overline{x}_n = X_n/n$—empirical average;
$R_n/S_n$—normalized range of accumulated sums $R_k$, $k \leq n$.

Based on the formula for Brownian motion, the system displacement (normalized range) Hurst proposed to calculate using the following relation:

$$R_n/S_n = (at)^H, \tag{4}$$

from where

$$H = \frac{\ln(R_n/S_n)}{\ln(at)}, \tag{5}$$

where $H$ is the Hurst exponent, varying from 0 to 1; $R_n/S_n$ is the rescaled range; $t$ is the studied period, and $a$ is a constant. The value of the coefficient $H$ characterizes the ratio of the strength of the trend (deterministic factor) to noise level (random factor). The indicator $H$ is a tool for determining the systems persistence behavior and gives an answer to the question of what the next value of the investigated series would be, more or less than the current value. The processes for which $H = 0.5$ have an independent data distribution, and are characterized by the absence of a trend (classical Brownian

motion). Time sequences for which *H* is greater than 0.5 are classified as persistent, preserving the existing trend. If the increments were positive for some time in the past—there was an increase—then on the average there will be an increase, this corresponds to a good predictability of the series. Thus, for a process with *H* > 0.5 there is a tendency to increase in the future, the effect of long-term memory is preserved. The case $0 < H < 0.5$ is characterized by antipersistence and is characterized by an alternating tendency.

On a large empirical material, it was found that the Hurst index value of the series of various natural processes is grouped in the interval *H* = 0.72–0.74. [38,41]. The question of why this is so remains open. Note that our average value of the Hurst index for time series of temperatures *H* = 0.74, also fell in this interval.

## 3. Analysis and Discussion

Using the method of spatial correlations, the spatial relationships between surface air temperature and precipitation at individual stations of the studied region were determined. This described the general spatial patterns of temperature and precipitation in the region, and generated a preliminary assessment of the features of regional climate formation.

Reference [45] describes that in all climatic zones of the region, changes in mean annual temperatures, unlike precipitation, were synchronous in time (Figure 2). In addition to the main climate-forming factors (radiation and circulation), climate of the Caucasus region is greatly influenced by the relief of the terrain, the orography of the terrain, and the distance of weather stations from each other. Figure 2b shows that changes in the precipitation regime in different climatic zones, unlike changes in the temperature regime (Figure 2a), were not synchronous.

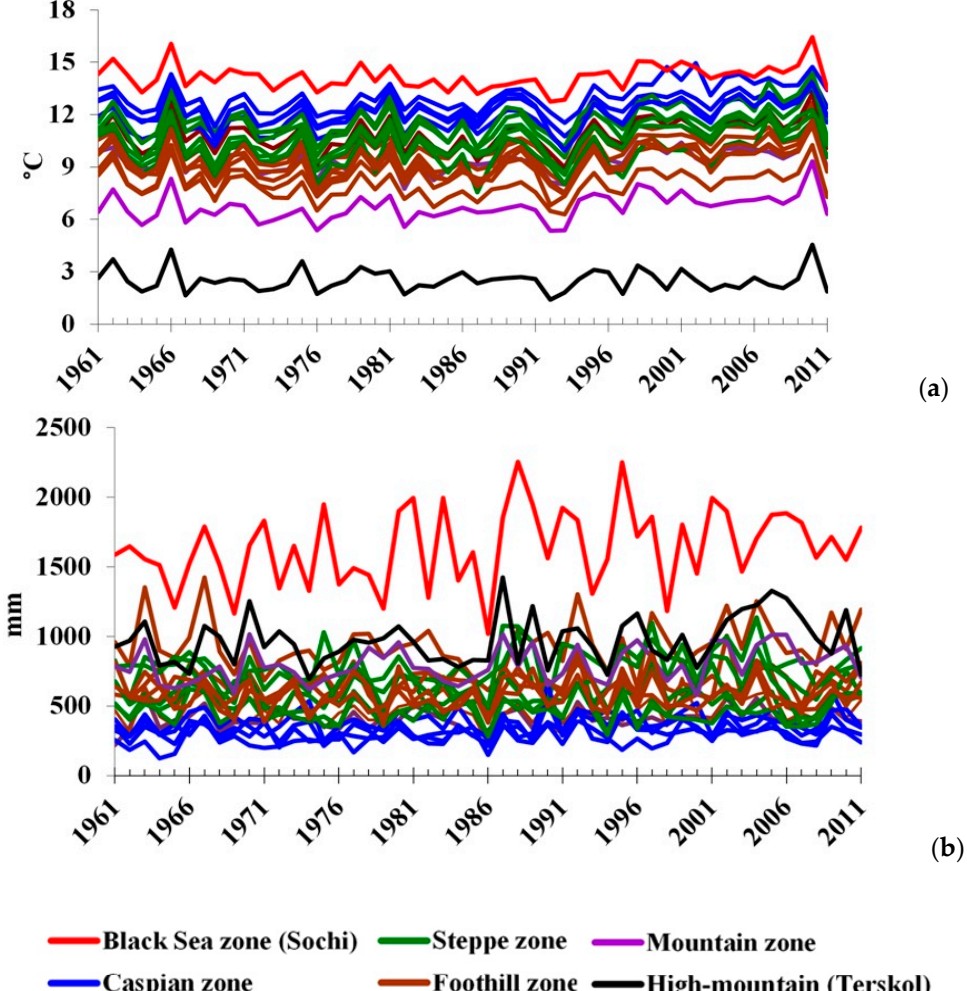

**Figure 2.** The course of average annual temperatures (**a**), and precipitation sums (**b**) according to data from 20 weather stations in southern Russia.

Apparently, this can be explained by influence of the same centers of low-frequency atmosphere variability, which by means of synoptic scale objects (cyclones, anticyclones, etc.), exert a distant influence on the climate in certain regions of Eurasia and are sources of anomalies in various meteorological fields.

The wide range of mean temperature dispersions and their rate of change in different climatic zones was also determined by the stations' location (from Rostov-on-Don in the north to Derbent in the south, and from Sochi in the west to Isberg in the east of the Caucasian region). Figure 2 show that the average annual temperature was maximum in Sochi (red line, $t_{av}$ = 14.18 °C), and minimum in Terskol (black line, $t_{av}$ = 2.5 °C). In all climatic zones, temperature changes occured synchronously, while temperature changes of stations within each climatic zone fit into their ranges.

The method of calculating spatial correlations was used to determine the spatial relationships of meteorological parameters (temperature, precipitation), measured at stations of different climatic zones of the southern Russia. Let $x_i$ (*j*) be the average annual air temperatures or the precipitation sums at some station *i*, where $j = 1, \ldots N_i$, *j* is the year, $N_i$ is the total number of measurements. Similar measurements at another station, *l*, respectively, $x_l$ (*k*), where $l = 1, \ldots N_k$, *i* is the year, $N_k$ is the total number of measurements. For joint analysis, it was necessary that $N_i = N_k$. The available series with average annual temperatures and precipitation sums of 20 stations satisfied this requirement.

The linear relationship between series $x_i$ (*j*) and $x_l$ (*k*) was quantitatively expressed by the Pearson correlation coefficient $R_{ik}$. Calculating all possible combinations of $R_{ik}$, we obtained matrices $A_c$,

where $c = T$ (temperature) or $c = P$ (precipitation), symmetric with respect to the main diagonal, which consisted of units (correlation matrices). The latter can be associated with the same matrix Z of distances between each pair of matrices. The distance ($L$) between stations was calculated based on the geographical coordinates (latitude $\varnothing$, longitude $\lambda$, Table 1) of each of the station pairs according to the formula:

$$L = \arccos(sin\varnothing 1 sin\varnothing 2 + cos\varnothing 1 cos\varnothing 2 cos\Delta\lambda) \tag{6}$$

Scatter diagrams of correlation matrices for air temperature and precipitation in the investigated region are presented in Figure 3. Each matrix elements was shown depending on the distance of the stations from each other, set in the distance matrices. Comparison of Figure 3a,b implied a significant difference in the correlation structure of the fields.

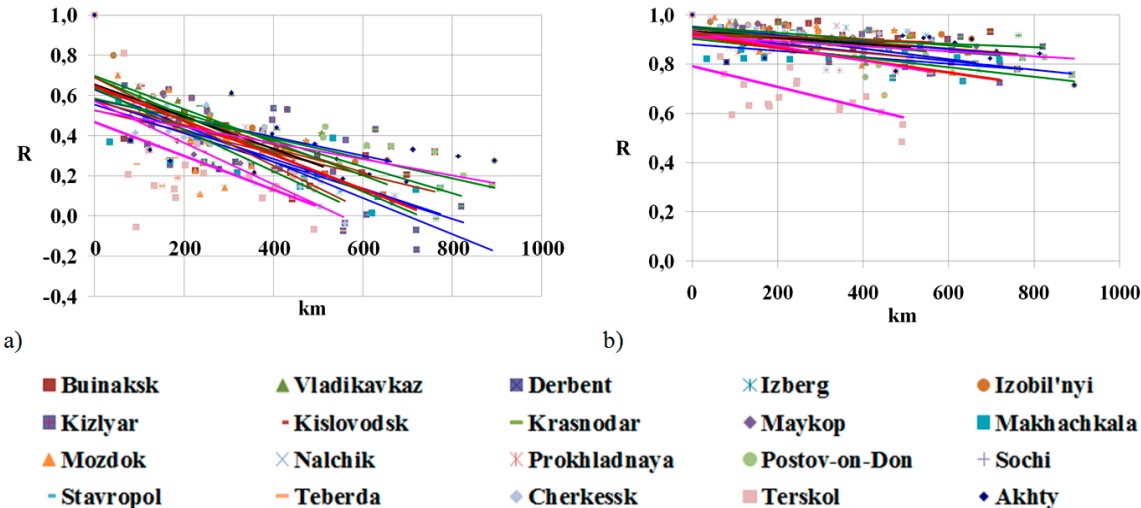

**Figure 3.** Spatial correlation of precipitation (**a**), and temperature (**b**) of the 20 weather stations, depending on the distance between them: the *x*-axis—distance between pairs of stations; and the *y*-axis—the correlations between the meteorological parameters (temperature, precipitation) of the 20 weather stations.

The spatial correlation of precipitation for all climatic zones tended to zero for scales of distances of the order of 600 km, and took negative values (from −0.04 to −0.2) for pairs of stations Derbent—Cherkessk, Derbent—Teberda, Derbent—Terskol, Derbent—Sochi, and Derbent—Maykop.

Such an inverse relationship between the precipitations of these stations can be explained, firstly, by the special geographical position of Derbent. Derbent is located on the western shore of the Caspian Sea (42.04° N; 48.17° E, 30 m a.s.l.), where the mountains of the Greater Caucasus are closest to the Caspian Sea, leaving only a narrow three-kilometer strip of plain, forming a semi-dry subtropical climate. Secondly, the great difference in altitude, and the remoteness of Derbent from other stations.

If we set the correlation threshold $R = 0.5$, it can be seen from Figure 3a that the correlation of precipitation decreased from $R = 0.7$ to $R = 0.5$ at a distance of 200 km, and it decreased further passing through $R = 0$ at a distance from 600 to 800 km.

The temperature correlations of different weather stations were quite high at distances from 0 to 1000 km, and lay mainly in a narrow range from $R = 1$ to $R = 0.7$ for plain, foothill, and mountain zones. The correlation between the temperatures of the Terskol (high-mountain zone) station with other stations was lower and varied from $R = 1$ to a threshold value $R = 0.50$ only for Terskol–Rostov-on-Don ($R = 0.554$) and Terskol–Derbent ($R = 0.485$).

Thus, the magnitude of temperature correlation depended mainly on the stations' location relative to height above sea level. In addition to this factor, the correlation of precipitation was determined by the geographical location of the weather stations (orography of the terrain) and the stations' remoteness from each other. The obtained values of the spatial correlation of temperature and precipitation

explained the synchronous course of temperature variation in all climatic zones, and the regional precipitation regime in all climatic zones of southern Russia (Figure 2.)

Various estimates of the change in global surface air temperature have been given [1–5,46,47]. From the second half of the 20th century, and in the first decade of the 21st century, the rate of temperature growth on average has varied in the range $0.17 \pm 0.01$ °C. According to our estimates, the warming trend across the Caucasus region corresponds to the overall trend of global temperature changes over the same period. Table 2 shows that the average annual temperature increased during the period 1961–2011 at 0.2 °C/10 years, while in 1961–1975 there was an insignificant negative trend, and since 1976 the temperature growth rate reached 0.43 °C/10 years with the trend contribution to the explained variance $D = 31.5\%$. The same trend was observed for all seasonal temperatures.

**Table 2.** Characteristics of seasonal and annual average air temperatures in the south of Russia.

| Temperature, °C | Winter | Spring | Summer | Autumn | Annual |
|---|---|---|---|---|---|
| Average temperature, 1961–2011 | −0.5 | 9.4 | 20.8 | 10.8 | 10.1 |
| Standard deviation, 1961–2011 | 1.5 | 0.9 | 1.0 | 1.1 | 0.8 |
| Anomalies, 1961–2011 | 0.3 | 0.1 | 0.4 | 0.1 | 0.2 |
| The angular coefficient of the trend (1961–2011), °C/10 years (*D*\*, %) | 0.22 (4.4%) | 0.08 (1.6%) | **0.33 \*\*** **(23%)** | 0.15 (4.1%) | **0.2** **(13%)** |
| The angular coefficient of the trend (1961–1975), °C/10 years (*D*, %) | −1.34 (9.2%) | 0.16 (0.5%) | 0.07 (0.1%) | 0.17 (0.7%) | −0.32 (3.3%) |
| The angular coefficient of the trend (1976–2011), °C/10 years (*D*, %) | 0.38 (9.3%) | 0.21 (6.1%) | 0.65 (41%) | 0.47 (18.4%) | 0.43 (31.5%) |

\* *D*, the trend contribution to the explained variance. \*\* statistically significant trends are marked in bold (*n* = 51).

Negative trend of the average winter temperature in 1961–1975 (−1.34 °C/10 years) since 1976 changed to a positive direction (+0.38 °C/10 years). Interannual variability of temperature from all seasons was greatest in the winter season, 1.5 °C. The main reason for the large winter variability was the large temperature difference in winter between low and high latitudes. Since 1976, the growth rate of all other seasonal temperatures increased. If in the period 1961–2011 the trend of seasonal temperatures was statistically significant only in the summer season (0.33 °C/10 years, *D* = 23%), in the modern period, trends in autumn (0.47 °C/10 years, *D* = 18.4%) and winter (0.38 °C/10 years, *D* = 9.3%) were added to statistically significant trends. The maximum value of seasonal anomalies ($\Delta T = 0.4$ °C) was also observed for summer temperatures. Since the mid 90s of the 20th century, there have been exceptionally positive anomalies in summer temperatures.

Next, we performed a comparative analysis of temperatures (average, maximum, minimum) within all climatic zones of the Caucasus region. The average values of mean annual temperatures, their standard deviation, characterizing the interannual variability, as well as the upper and lower boundaries of the intervals of mean annual air temperature in different climatic zones (at 95% confidence intervals) are presented in Table 3. The lower and upper limits of the confidence intervals of the mean annual temperature, according to data from the Black Sea, Caspian, steppe, piedmont, and mountain weather stations, intersect. The average annual temperature in Terskol (2.5 °C, taking into account the interannual variability from 1.22 °C to 3.78 °C) was significantly lower than the rest, which was explained by the high altitude zoning. This station was also characterized by the stability of the change in annual temperature (0.01 °C/10 years). In our study, a high-altitude zone was distinguished (Terskol, 2144 m a.s.l.) with a negative trend of average annual temperature for the period 1961–2011 (−0.01 °C/10 years), and for the years 1961–1975 (−0.01 °C/10 years). Since 1976, the negative trend changed its direction to positive (+0.05 °C/10 years).

**Table 3.** Characteristics of the temperature regime of surface air in the different climatic zones of southern Russia.

| Temperature | Black Sea Zone (Sochi) | Steppe Zone | Caspian Zone | Foothill Zone | Mountain Zone | High-Mountain (Terskol) |
|---|---|---|---|---|---|---|
| Average temperature $t_{av}$ ($\sigma$), °C | | | | | | |
| Annual temperature (st. deviation) | 14.18 (0.72) | 10.82 (0.92) | 12.38 (0.82) | 9.24 (0.91) | 8.05 (0.78) | 2.5 (0.64) |
| Upper bound * | 15.62 | 12.67 | 14.02 | 11.05 | 9.6 | 3.78 |
| Lower bound | 12.74 | 8.98 | 10.73 | 7.43 | 6.49 | 1.22 |
| Rates of change of average temperature, °C/10 year (*D*, %) | | | | | | |
| Annual | | | | | | |
| (a) 1961–2011 | 0.06 (2%) | **0.25 (14%)** | **0.21 (17%)** | **0.23 (17%)** | **0.17 (11%)** | −0.01 (0%) |
| (b) 1961–1975 | −0.3 (4%) | −0.03 (2%) | −0.37 (5%) | −0.45 (7%) | −0.5 (6%) | −0.2 (2%) |
| (c) 1976–2011 | 0.31 (19.6%) | 0.48 (28%) | 0.42 (30%) | 0.5 (33%) | 0.38 (24.7%) | 0.05 (0.7%) |
| Winter | −0.04 (0.2%) | 0.3 (6%) | 0.18 (4%) | 0.27 (6%) | 0.14 (2%) | −0.02 (0%) |
| Spring | −0.03 (0.2%) | 0.09 (2%) | 0.11 (4%) | 0.1 (2%) | 0.06 (1%) | −0.07 (1%) |
| Summer | 0.28 (19%) | 0.34 (20%) | 0.26 (18%) | 0.37 (22%) | 0.35 (30%) | 0.29 (25%) |
| Autumn | 0.07 (1%) | 0.16 (4%) | 0.15 (4%) | 0.19 (6%) | 0.13 (3%) | −0.07 (1%) |

* the upper (lower) boundary of the mean temperature ($t_{mean} \pm 2\,\sigma$) at 95% confidence interval; ** statistically significant trends are marked in bold (*n* = 51).

Changes in the average annual air temperature in different climatic zones of southern Russia were also represented by three periods: 1961–2011, 1961–1975, and 1976–2011 in Table 3. During the period 1961–2011 in all climatic zones of southern Russia, with the exception of the Black Sea zone (Sochi) and the high-mountain zone (Terskol), an average annual temperature increased from 0.17 °C/10 years in the mountain zone to 0.26 °C/10 years in the steppe zone. In the Black Sea zone, the rate of change in the mean annual temperature was 0.06 °C/10 years, and 0.01 °C/10 years in the high-mountain zone, which characterizes a stable temperature regime in these zones. According to the station Makhachkala (the Caspian region), it also received insignificant rates of change in the average annual temperature by 0.08 °C/10 years. The stability of average annual temperatures (0.08 °C/10 years) in Makhachkala was observed against the background of an increase in absolute maximums of temperatures, and an equally significant decrease in absolute minimums of temperatures.

It is probably due to regional features of the terrain: large water bodies (Black Sea, Caspian Sea) and snow massifs in the high-mountain zone that smooth out the amplitude of the change in the mean annual temperature.

During periods of seasonal average temperatures in all climatic conditions from 1961 to 2011, a stable pattern of temperature growth rates was observed in summer seasons: from 0.26 °C/10 years (*D* = 18%) in the Caspian zone to 0.37 °C/10 years (*D* = 22%) in the foothill zone.

The results of calculations of the Hurst exponent for determining the trend-persistence of precipitation and temperature series are presented in Table 4. Table 4 shows that the persistence indicators for temperature trends significantly exceeded the values for precipitation, and it characterizes the persistence and long-term changes in the temperature regime. The highest persistence trends have been observed at average annual, summer temperatures (*H* = 0.80), and autumn temperatures (*H* = 0.73). The annual, summer (*H* = 0.75), and autumn maximums (*H* = 0.70), in addition to spring minimum temperatures (*H* = 0.72) have had persistent trends. Since fractality is associated with determinism [41], it can be assumed that the summer warming observed in recent decades is a consequence of the coordinated influence of a number of climate-forming factors.

**Table 4.** Hurst exponent for climatic characteristics according to the data from 20 weather stations in the southern Russia.

| Meteoparameters | Hurst Exponent (Standard Deviation), $H$ ($\delta$) | | | | |
|---|---|---|---|---|---|
| | Winter | Spring | Summer | Autumn | Annual |
| 1 | 2 | 3 | 4 | 5 | 6 |
| Precipitation total | 0.63 (0.06) | 0.62 (0.07) | 0.61 (0.07) | **0.68 (0.06)** | 0.64 (0.09) |
| Daily maximums of precipitation | 0.62 (0.07) | 0.59 (0.08) | **0.63 (0.06)** | **0.63 (0.07)** | 0.62 (0.06) |
| Average temperature | 0.70 (0.05) | 0.64 (0.08) | **0.80 (0.05)** | 0.73 (0.05) | 0.80 (0.06) |
| Maximum temperature | 0.66 (0.07) | 0.64 (0.06) | **0.75 (0.05)** | 0.70 (0.08) | 0.75 (0.04) |
| Minimum temperature | 0.68 (0.08) | **0.72 (0.08)** | 0.68 (0.09) | 0.61 (0.08) | 0.67 (0.08) |

Previous studies [31] have found that, with the exception of the steppe and Black Sea zones, seasonal increase in precipitation amounts prevail. At the same time, an autumn increase in precipitation amounts was observed at all stations without exception. For annual sums of precipitation, in 27% of the steppe stations studied and in 10% of foothill stations, they decreased, for all other meteorological stations an increase was observed in annual sums of precipitation. In this case, Table 4 shows that in the autumn, for all stations the highest Hurst exponent value was observed both for the sum of precipitation and for the daily maximums $H = 0.63$, which characterized the persistence of the identified trends for a long period.

We used the data from Tables 3 and 4 to visualize changes in summer temperature regimes with indicators characterizing the statistical stability and persistence of the detected changes. Figure 4 shows the rates of change in the mean annual temperature (°C/10 years) with the coefficient of determination ($D$, %) for different climatic zones, against the background of the distribution fields of the Hurst exponent ($H$). It is seen from Figure 4 that the increase in average summer temperatures is not only distinguished by its high values in all climatic zones, but by the highest Hurst coefficients that characterize the persistence of the obtained trends. In the summer season, Hurst's exponent had a small spread of values from $H = 0.79$ in the steppe and Caspian zones, to $H = 0.82$ in the foothill, mountainous areas, from which a steady increase in average summer temperatures should be expected. The highest Hurst exponents characterized the process of growth of these temperatures as persistent, having a long-term memory, with a high probability of continuing in the future.

Thus for average, maximum, and minimum temperatures, the Hurst exponent lies in the range $0.5 < H < 1$, and all processes belong to the class of persistent ones that preserve the existing trends. Such a long-term memory takes place regardless of time scale. All annual changes are correlated with all future annual changes. The existing trend of rising surface air temperatures will continue in the future, at least for the next 50 years (for the period analyzed using the rescaled range method). Since fractality is related to determinism [34,36], it can be assumed that the increase in temperature observed in recent decades is a consequence of the coordinated effect of a variety of climate-forming factors, both natural and anthropogenic.

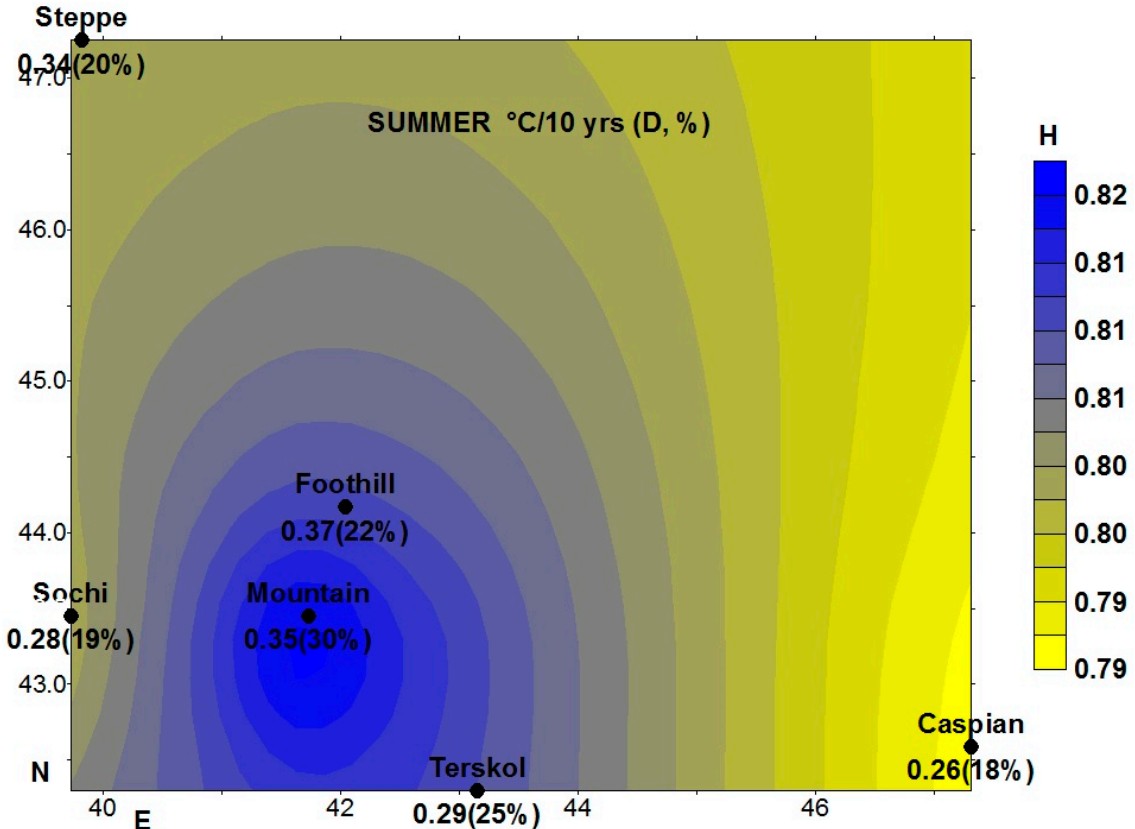

**Figure 4.** The rates of change in the average temperature (°C/10 years) with the coefficients of determination (*D*, %) in the summer season for different climatic zones, against the background of the trend-persistence field (the Hurst exponent *H* in color).

## 4. Conclusions

This study allowed us to approach the question different climatic zones sensitivity (plains, foothills, mountains) to climate change, and whether warming differs in mountains vs. plains. This is important because changes in temperature and precipitation in the mountains lead to significant changes in the hydrological cycle, in particular, to the observed process of deglaciation of the Caucasus glaciers. Furthermore, climate change in the areas of intensive farming in the south of Russia (plain, foothill) must be taken into account when solving problems in agriculture.

According to our study results, and taking into account previous studies of climate change in the Caucasus region, it became clear that the warming tendency prevails in all climatic zones with some features. Changes in the precipitation regime are not so unidirectional.

1. In the period 1961–2011, the average annual temperature in the entire territory of the southern Russia increased by 0.2 °C/10 years (*D* = 13%) with the most persistent trend being during the summer season (0.33 °C/10 years, *D* = 23%).

2. In all climatic zones of southern Russia, with the exception of the Black Sea zone (Sochi) and the high-mountain (Terskol), average annual and season temperatures have increased during 50 years of observation. In Sochi and Terskol, a statistically significant increase was observed only at average summer temperatures.

Since the beginning of global warming (since 1976), there has been the significant increase in the growth of average, maximum, and minimum temperatures in all climatic zones.

3. A change in the precipitation regime does not manifest itself as clearly as changes in temperature. During all seasons, increase and decrease in seasonal precipitation amounts occurred, but statistically insignificant. In all climatic zones, an increase in the amount of precipitation was observed in the autumn season, which was statistically significant in the steppe region.

4. Based on the study of the fractal properties of time series of precipitation and air temperature in the surface layer of the atmosphere in all climatic zones of southern Russia, it is shown that the Hurst exponent of the temperature ($H = 0.74$) trend significantly exceeds the Hurst exponent of the trend of precipitation ($H = 0.63$). It characterizes the persistence and long-term changes in the temperature regime. Of these, the trends of the average annual, summer ($H = 0.80$), and autumn ($H = 0.73$) temperatures are allocated. The changes in the maximum annual, summer ($H = 0.75$) and autumn ($H = 0.70$) temperatures, as well as minimum temperatures in spring ($H = 0.72$. High Hurst performance characterizes the process of increasing temperatures as constant, in contrast to increasing precipitation, having a long-term memory and with a high probability of continuation in the future.

5. Spatial correlation analysis performed for mean temperatures and precipitation sums of all climatic zones showed a high correlation ($R = 1.0$–$0.7$) between average temperatures of different climatic zones at distances of 0–1000 km between stations, and a decrease in correlation from 1 to 0.5 between precipitation sums at a scale of distances from 0 to 200 km.

**Author Contributions:** A.A.T. have formulated the statement of the problem and the main directions of research. Conclusions on the results of research are formulated. The final version of the article was edited before publication. B.A.A. have developed the algorithm necessary to solve the task. The analysis of the dynamics of changes in the temperature and precipitation regime was carried out. The first version of the article was formed before publication. L.A.K. time series of meteorological parameters were computed by methods of mathematical statistics. The obtained statistics of the meteorological parameters are formed into tables, their description is given. N.V.T. an array of data was prepared, a check was made for the homogeneity of time series. Graphs with linear trends, graphs with anomalies of meteoparameters are constructed.

**Funding:** This research received no external funding.

**Acknowledgments:** The authors are grateful to the reviewers for comments on the work, which led to a better presentation of the material.

**Conflicts of Interest:** The authors declare no conflicts of interest.

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
