# Peer review of "Analysis of Climate Change in the Caucasus Region: End of the 20th–Beginning of the 21st Century"

_climate, doi:10.3390/cli7010011_

Round 1

Reviewer 1 Report

I consider work valuable for publishing.

The authors presented the research in an innovative manner, emphasizing the importance of linking regional climate change to global ones.

Please Switch to Latin:

Для калибровки временных измерений Херст ввел безразмерное отношение посредством деления размаха R на стандартное отклонение S наблюдений. Размах Rn является разностью между максимальным и минимальным уровнями накопленного отклонения Xn. 

Author Response

Response to Reviewer 1(Round 3)  Comments

We would like to thank the reviewer for the productive dialogue with the authors aimed at improving  the  paper.

Point:The authors presented the research in an innovative manner, emphasizing the importance of linking regional climate change to global ones.

Please Switch to Latin:

Для калибровки временных измерений Херст ввел безразмерное отношение посредством деления размаха R на стандартное отклонение S наблюдений. Размах Rn является разностью между максимальным и минимальным уровнями накопленного отклонения Xn. 

Response: Thanks for the comment, we accidentally left a paragraph in Russian, which is translated below into English. The paragraph in Russian has been deleted.

Reviewer 2 Report

I would like to thank the authors for their efforts made to improve the manuscript.
The only three points I would like to highlight are:
1) page 6 - a part already translated into English still appears in Russian - just to be deleted
2) table 3 - still contains: absolute maximum/minimum. Please let me just briefly explain ... In climatology we use absolute values to present the ONLY values for maximum and minimum that happened during a period. There can be only one absolute minimum and one absolute minimum for a period 1961-2011 for each month or season. If you wish to highligt the change/trend in these highest/lowest values you can use any method to define extreme values and then check the rate of change during the period ... As a climatologist I cannot accept using the expression: Rates of change of absolute maximum/minimum
3) Fig 2 - please put letters A and B in such a place which will be well seen (the legend should go under the whole); references - please go through and check if they are all compiled according to the instructions (e.g. 41)

Author Response

Response to Reviewer 2(Round 3)  Comments

We would like to thank the reviewer for the productive dialogue with the authors aimed at improving  the  paper.

Point 1: page 6 - a part already translated into English still appears in Russian - just to be deleted.

 Response 1:  The 2nd paragraph in Russian has been deleted .

Point 2:  table 3 - still contains: absolute maximum/minimum. Please let me just briefly explain ... In climatology we use absolute values to present the ONLY values for maximum and minimum that happened during a period. There can be only one absolute minimum and one absolute minimum for a period 1961-2011 for each month or season. If you wish to highligt the change/trend in these highest/lowest values you can use any method to define extreme values and then check the rate of change during the period ... As a climatologist I cannot accept using the expression: Rates of change of absolute maximum/minimum/

Response 2:  Thank you for the constructive dialogue with us. We agree with your arguments and remove the trend data of the absolute maximums and minimum temperatures from Table 3 and their analysis from the text of the article.

3) Fig 2 - please put letters A and B in such a place which will be well seen (the legend should go under the whole); references - please go through and check if they are all compiled according to the instructions (e.g. 41)

Response 3:    We took out the letters a) and b) outside the picture field to improve the perception of Figure 2. 

Reviewer 3 Report

The study includes an analysis of temperature (mean, maximum and minimum) and precipitation trends and persistence for the region of Caucasus based on 1961-2011 ground station data.  

A novel element of this study is the incorporation of the Hurst exponent in characterising the trend persistence, which indicates fractality and determinism. I found this interesting but some more explanation can be given to why a high H number (such as the one calculated for annual and summer temperatures) may suggest continuation of the respective trend in the future. I was confused also because the authors cite a study that this concept was used in financial markets [40] which we know exhibit chaotic behaviour and not deterministic. Perhaps an example from geophysics time-series analysis and prediction might help.

The authors need to clarify if the figures presented in this manuscript are taken from previous publications (see, for example, my comment for Fig. 2 in page 8). A more clear set of statements must be formulated in order to define the motivation for analysing the Caucasus region (see also specific comment below on moving text from section 2 to the Intro).

The conclusions about the observed trend over the high-mountain area cannot be generalised since are based on the results of only one station (Terskol).  Perhaps the effect of elevation would be interesting to be explored by looking at the potential association of the observed trend vs altitude from all 20 stations. For example, plot trends vs altitude for the 0 data points stations (e.g. for temperature).

The respective trend calculation results for precipitation are not shown.

Other specific comments:

page 2, first para: You must provide some more quantitative info on the periods (base and comparison one) that the Russian and global anomalies refer to.

pages 2-3: at least the first three paragraphs of section 2 refer to background info on the Caucasus region, which is the object of this study, therefore must be moved in the introduction.

page 3, past para: "... in the study area were added [31,32]". As datasets or as conclusions? And, where "were added"?

page 6, first para: " British hydrologist Harold Hearst". The surname is "Hurst".

page 6: remove (or translate) the 2nd paragraph in Russian

page 7,last para: "H= 0.72 ÷ 0.74". The symbol "÷" usually depicts fraction or division. here for the range, a "-" can be used.

page 8, section title: it should be "3. Analysis and Discussion"

page 8, 2nd para: "It was published in [47] that in all climatic zones of the region the changes in mean annual temperatures, unlike precipitation, are synchronous in time (Fig.2)." So, Figure 2 is taken from Reference [47]?

page 10, Fig 3: The increasing correlation (among pairs of stations) with their decreasing distance is trivial. It just confirms that stations close to each other have similar climate and variability.

page 13, 3rd para: "In the dynamics of seasonal average temperatures in all climatic zones from 1961 to 2011 there was a general pattern - stable temperature growth rate in summer seasons: from 0.26 °C/10 years (D = 18%) in the Caspian zone to 0.37 °C/10 years (D = 22%) in the foothill zone". You might want to rephrase it to: "consistent pattern of temperature growth rate in summer seasons"

Reference [5] is about "Climate: Observation. Projections and Impacts (Russia)" but it is connected with a web link (of CRU) which I do not get the relevance: http://www.cru.uea.ac.uk/cru/

In term of English, the articles are missing at the start of sentences.

Author Response

Response to Reviewer 3(Round 3)  Comments

We would like to thank the reviewer for the productive dialogue with the authors aimed at improving  the  paper.

Point 1: page 6 - a part already translated into English still appears in Russian - just to be deleted.

 Response 1:  The 2nd paragraph in Russian has been deleted .

Point 2:  table 3 - still contains: absolute maximum/minimum. Please let me just briefly explain ... In climatology we use absolute values to present the ONLY values for maximum and minimum that happened during a period. There can be only one absolute minimum and one absolute minimum for a period 1961-2011 for each month or season. If you wish to highligt the change/trend in these highest/lowest values you can use any method to define extreme values and then check the rate of change during the period ... As a climatologist I cannot accept using the expression: Rates of change of absolute maximum/minimum/

Response 2:  Thank you for the constructive dialogue with us. We agree with your arguments and remove the trend data of the absolute maximums and minimum temperatures from Table 3 and their analysis from the text of the article.

3) Fig 2 - please put letters A and B in such a place which will be well seen (the legend should go under the whole); references - please go through and check if they are all compiled according to the instructions (e.g. 41)

Response 3:    We took out the letters a) and b) outside the picture field to improve the perception of Figure 2. 

The study includes an analysis of temperature (mean, maximum and minimum) and precipitation trends and persistence for the region of Caucasus based on 1961-2011 ground station data.  

Point1: A novel element of this study is the incorporation of the Hurst exponent in characterising the trend persistence, which indicates fractality and determinism. I found this interesting but some more explanation can be given to why a high H number (such as the one calculated for annual and summer temperatures) may suggest continuation of the respective trend in the future. I was confused also because the authors cite a study that this concept was used in financial markets [40] which we know exhibit chaotic behaviour and not deterministic. Perhaps an example from geophysics time-series analysis and prediction might help.

Response1: We put a link to the source where fractal analysis (the rescaled range method)  is used in nature. In the «Materials and Methods», we noted that for the first time the rescaled range method was used in the study of the natural time series by Harold Hurst (link available).

Point2:The authors need to clarify if the figures presented in this manuscript are taken from previous publications (see, for example, my comment for Fig. 2 in page 8). A more clear set of statements must be formulated in order to define the motivation for analysing the Caucasus region (see also specific comment below on moving text from section 2 to the Intro).
Response2: Figure 2 is taken from our paper for clarity [47]. The figure shows the course of the average annual temperature and annual precipitation amounts according to 20 stations on the nerritory of the  southern Russia. We uses the same annual data, as well as added seasonal data, which are analyzed not for individual stations, but for different climatic zones in this article.

Point3:The conclusions about the observed trend over the high-mountain area cannot be generalised since are based on the results of only one station (Terskol).  Perhaps the effect of elevation would be interesting to be explored by looking at the potential association of the observed trend vs altitude from all 20 stations. For example, plot trends vs altitude for the 0 data points stations (e.g. for temperature).The respective trend calculation results for precipitation are not shown.

Response3: Yes, we agree that it is not correct to draw conclusions about the trends in the high-mountain zone for one station Terskol. But we briefly discussed the difficulties of obtaining data in high-altitude areas, and we decided to use the instrumental data even for one station. As for identifying the relationship of the observed trend with altitude at all 20 stations, this is seen in Figure 2, where the annual changes in meteorological parameters for each station are presented and the station indicates the station's belonging to the climatic zone (and in Table 1 - the distribution of zones in height). In more detail this topic can be developed in another study.

Other specific comments:

Point 4: page 2, first para: You must provide some more quantitative info on the periods (base and comparison one) that the Russian and global anomalies refer to.

Response4  We presented the rates of temperature change in Russia and on the globe in the modern period (1976-2017).

Point 5: pages 2-3: at least the first three paragraphs of section 2 refer to background info on the Caucasus region, which is the object of this study, therefore must be moved in the introduction.
Response5 We have transferred information about the Caucasus region in the «Introduction».

Point 6: page 3, past para: "... in the study area were added [31,32]". As datasets or as conclusions? And, where "were added"?

Response6  We changed the sentence to the following: «In previous studies [31, 32], trends in the amount of precipitation and daily maximums of precipitation were analyzed in the Caucasus region, an analysis of the temperature regime was added in this article».

Point 7: page 6, first para: " British hydrologist Harold Hearst". The surname is "Hurst".

Response7 We have corrected «Hurst»

Point 8: page 6: remove (or translate) the 2nd paragraph in Russian

Response8:  the 2nd paragraph in Russian have been  removed

Point 9: page 7, last para: "H= 0.72 ÷ 0.74". The symbol "÷" usually depicts fraction or division. here for the range, a "-" can be used.

Response9   We have corrected : «H = 0,72 - 0,74»

Point 10: page 8, section title: it should be "3. Analysis and Discussion"

Response10  We have corrected : "3. Analysis and Discussion"

Point 11: page 8, 2nd para: "It was published in [47] that in all climatic zones of the region the changes in mean annual temperatures, unlike precipitation, are synchronous in time (Fig.2)." So, Figure 2 is taken from Reference [47]?

Response11   Yes, Figure 2 is taken from our article for clarity [47].

Point 12: page 10, Fig 3: The increasing correlation (among pairs of stations) with their decreasing distance is trivial. It just confirms that stations close to each other have similar climate and variability.

Response12  Not certainly in that way. This shows the similarity of the temperature regime, and the similarity in the direction of change in temperature — increases or decreases in a given year, but the absolute values of temperature in different climatic zones differ significantly (especially the Black Sea and high-mountain zones). The precipitation regime for each climatic zone, as can be seen from Fig. 2 and 3, is completely autonomous.

Point 13: page 13, 3rd para: "In the dynamics of seasonal average temperatures in all climatic zones from 1961 to 2011 there was a general pattern - stable temperature growth rate in summer seasons: from 0.26 °C/10 years (D = 18%) in the Caspian zone to 0.37 °C/10 years (D = 22%) in the foothill zone". You might want to rephrase it to: "consistent pattern of temperature growth rate in summer seasons"

Response13  yes, thanks, we changed the wording:
“In the dynamics of seasonal average temperatures in all climatic zones from 1961 to 2011 there was a steady pattern of temperature growth rates in summer seasons: from 0.26 ° C / 10 years (D = 18%) in the Caspian zone to 0.37 ° C / 10 years (D = 22%) in the foothill zone ".

Point 14: Reference [5] is about "Climate: Observation. Projections and Impacts (Russia)" but it is connected with a web link (of CRU) which I do not get the relevance:
http://www.cru.uea.ac.uk/cru/

Response14  We have corrected the link